# Dynamic Analysis of a Pest Management Smith Model with Impulsive State Feedback Control and Continuous Delay

**Zhenzhen Shi, Yaning Li and Huidong Cheng ***

College of Mathematics and Systems Science, Shandong University of Science and Technology, Qingdao 266590, China

**\*** Correspondence: chd900517@sdust.edu.cn

**Abstract:** In our paper, we propose a single population Smith model with continuous delay and impulsive state feedback control. The application in pest management of this model is investigated. First, the singularity of this model is qualitatively analyzed; then, we consider the existence and uniqueness of order-one periodic orbit in order to determine the frequency of the implementation of chemical control. Moreover, based on the limit method of the sequences of subsequent points, we verify the stability of periodic orbit to ensure a certain robustness of this control; at last, we carry out the numerical simulations to verify the correctness of the theoretical results.

**Keywords:** unilateral diffusion; order-one periodic orbit; qualitatively analysis; Smith model

## 1. Introduction

As an important ecosystem, forests are of great significance in regulating the ecological environment [1,2]. However, rapid changes in climate and ecology have led to the decline and death of the forest ecosystem in recent years, such as, drought, wild fires, pathogens and insect pests, and so on. According to new research, the climate change like global warming leads to the explosion of the number of insects, and then the forest ecosystem will be more severely damaged [3]. Therefore, pathogens and forest insects are the most important pervasive disturbance in forest ecosystem. For example, the stem borer, whitefly and tobacco aphid caused substantial damage to the yield and quality of tobacco in India, and in British Columbia the deteriorating climate caused the outbreak of the mountain pine beetles. Thus, the control and management of the insects attracts the attention of many scholars, and some effective results are obtained [4–11].

In recent years, impulsive differential equations are widely used in practical issues [12–17] such as the pulses vaccination in diseases [18–22], pulse injection nutrient in the medium [23–26] and population dynamics [27–31]. Results suggest that the predator-prey system can be constructed by using impulsive differential equations to simulate the pest control and get some positive theoretical results [32–39], as exemplified in the release of predators periodically [40–42], the periodic release of pests that are infected with disease [43–45] and periodically release of predators combined with periodic sprayed pesticides [46,47]. However, scholars find that the impulsive control strategy on the basis of the density of insect pests population has more practical significance. Zhao et al. [48] proposed a predator-prey model with impulsive state-dependent feedback control, and verified that the stable periodic solution exists in the model. Zhang et al. [1] investigated a predator-prey Gompertz model with two state-dependent impulses, and proved the existence and stability of periodic solution. Sun et al. [49] considered a pest management predator–prey system and formulated an optimization issue to get the optimum insect control level. Nevertheless, these studies have mainly been mainly

used to investigate the application of multi-population systems in pest management, and few scholars research single population systems in pest management [50].

Logistic model,

$$\frac{dx}{dt} = rx\left(1 - \frac{x}{K}\right),$$  (1)

is a simple and important ecological system in biomathematics, where $r$ is defined as a rate of increase with unlimited food, $K$ is the value of $x(t)$ at saturation. System (1) is based on the assumption that the relative growth rate $\frac{dx}{dt}$ of the population size is linear function $1 - \frac{x}{K}$. Therefore, some scholars believe that, to some extent, Logistic equations confirm the logic of population growth and finite resources. But it is not a knowledge of biological individual regeneration or nutrient supply. The model is mainly suitable for low-level biological populations such as bacteria, yeast, and planktonic algae [51]. In 1963, Smith [52] studied an algae named Daphnia in the laboratory and found that the data of this population did not conform to the linear law. Therefore, Smith assumed that the relative growth rate of the population size at time $t$ was proportional to the amount of food remaining at that time, not the total amount of food. That is,

$$\frac{1}{x}\frac{dx}{dt} = r\left(1 - \frac{G(t)}{T}\right),$$  (2)

where $G$ is consumption rate, $T$ is the demand for food when the population reaches saturation. $r$ is the intrinsic growth rate. Smith assumed that the food consumed by the population is mainly to maintain the food $c_1 x(t)$ that the organism needs and the food $c_2\frac{dx}{dt}$ it needs to breed. Then,

$$G(x) = c_1(x) + \frac{dx}{dt}.$$  (3)

Therefore,

$$\frac{dx}{dt} = rx\left(1 - \frac{T - c_1 x}{T + c_2 x}\right).$$  (4)

When the population size reaches the maximum capacity $K$ of the environment, then $\frac{dx}{dt} = 0$, that is, the population no longer grows, then the food is only used for survival needs: $T = G(t) = c_1 K$. $T = c_1 K$ is substituted into (4) and set $c = \frac{c_2}{c_1}$. Thus we get the following model

$$\frac{dx}{dt} = rx\left(\frac{K - x}{K + (\frac{r}{c})x}\right).$$  (5)

Because the growth rate of the pest population depends not only on the density of the time $t$, but also on the population density of all the times in the past, we call this is continuous delay. Thus we discuss a single population Smith model with continuous delay and state-dependent impulsive feedback control as follows,

$$\begin{cases} x'(t) = x(t)\left(\frac{r(K - x(t))}{K + Dx(t)} - b\int_{-\infty}^{t} e^{-a(t-s)}x(s)ds\right), \ x < h, \\ \triangle x(t) = -vx(t), \ x = h, \end{cases}$$  (6)

where $\triangle x(t) = x^+(t) - x(t)$, i.e., $\triangle x(t)$ represents the change in population density of pests, $x(t)$ is defined as the density of pest population at time $t$, $v$ is proportion of killing pests by spraying insecticides, and $h$ denotes the economic threshold of pests, i.e., the control measure is adopted when pest population density reaches $h$.

Let $y(t) = \int_{-\infty}^{t} e^{-a(t-s)} x(s) ds$, then

$$
\begin{aligned}
\frac{dy(t)}{dt} &= e^{-at} e^{at} x(t) + -ae^{-at} \int_{-\infty}^{t} e^{as} x(s) ds \\
&= x(t) + -a \int_{-\infty}^{t} e^{-a(t-s)} x(s) ds \\
&= x(t) - ay(t),
\end{aligned}
$$

since the variable $y$ has no direct ecological significance. Therefore, model (6) is converted to model (7)

$$
\left.
\begin{cases}
\left.
\begin{aligned}
\frac{dx(t)}{dt} &= x(t) \left( \frac{r(K - x(t))}{K + Dx(t)} - by(t) \right), \\
\frac{dy(t)}{dt} &= x(t) - ay(t),
\end{aligned}
\right\} \quad x < h, \\
\left.
\begin{aligned}
\triangle x(t) &= -vx(t), \\
\triangle y(t) &= 0,
\end{aligned}
\right\} \quad x = h.
\end{cases}
\right.
\tag{7}
$$

Assume $D = \frac{r}{c}$, the parameters of this system are positive and $\int_{0}^{\infty} e^{-s} ds = 1$.

The rest of our paper is organized as follows: In the next section, we analyze qualitatively the singularity of model, and discuss the existence and uniqueness of periodic solution of model (7) by means of subsequent functions, and the theory of Bendixson-Dulac theorem and the limit method of subsequent points sequences which proved the stability of periodic orbit. At last, we carried out the numerical simulations to verify the correctness of the theoretical results.

## 2. Dynamic Analysis of Impulsive State Feedback Control Model

In this section, the qualitative analysis of the model (7) is first performed, and then the existence, uniqueness and stability of the order-one periodic orbit of model (7) are proved by means of successor function and the geometry theory of impulsive differential equation. For convenience, we denote the order-one periodic orbit as OOPO.

### 2.1. Qualitative Analysis of Pest Management Model without Impulsive

In this section, we will consider system (7) without pulse, that is to say, $v = 0$. Then, we obtain a system (8) as follows

$$
\begin{cases}
x'(t) = x(t)[\frac{r(K - x(t))}{K + Dx(t)} - by(t)], \\
y'(t) = x(t) - ay(t).
\end{cases}
\tag{8}
$$

Then, we solve the functions

$$
\begin{cases}
x(t)[\frac{r(K - x(t))}{K + Dx(t)} - by(t)] = 0, \\
x(t) - ay(t) = 0.
\end{cases}
\tag{9}
$$

which has two equilibria: $O(0,0)$ and $E^*(x^*, y^*)$ where

$$
x^* = \frac{\sqrt{\Delta} - (ar + bK)}{2bD},
$$

$$
y^* = \frac{\sqrt{\Delta} - (ar + bK)}{2abD},
$$

and $\Delta = (ar + bK)^2 + 4DKrab$.

In the rest of this article, we let $(H)$ represent condition $K < x^* \sqrt{D}$.

The Jacobian matrix of system (7) is as follows

$$
J = \begin{pmatrix} \dfrac{r(K^2 - 2Kx - Dx^2)}{(K + Dx)^2} - by & -bx \\ 1 & -a \end{pmatrix}.
$$

At $O(0,0)$, we have

$$
J(O) = \begin{pmatrix} r & 0 \\ 1 & -a \end{pmatrix},
$$

thus, when $(H)$ holds, $O(0,0)$ is a saddle point.

At $E^*(x^*, y^*)$, we have

$$
J(E^*) = \begin{pmatrix} \dfrac{r(K^2 - 2Kx^* - Dx^{*2})}{(K + Dx^*)^2} - by^* & -bx^* \\ 1 & -a \end{pmatrix},
$$

then, we have

$$
tr(J(E^*)) = \frac{r(K^2 - 2Kx^* - Dx^{*2})}{(K + Dx^*)^2} - by^* - a,
$$

$$
det(J(E^*)) = \frac{-ar(K^2 - 2Kx^* - Dx^{*2})}{(K + Dx^*)^2} + aby^* + bx^*.
$$

If $(H)$ holds, then $tr(J(E^*)) < 0$ and $det(J(E^*)) > 0$. That is to say, $E^*$ is a locally asymptotically stable node or focus. Then we get the following theorem.

**Theorem 1.** *If $(H)$ holds, then system (7) has a globally asymptotically stable equilibrium point $E^*$.*

**Proof.** Let Dulac function $B = \frac{1}{x}$, then we have

$$
D = \frac{\partial(PB)}{\partial x} + \frac{\partial(QB)}{\partial y} = -\frac{rK(1 + D)}{(K + Dx)^2} - \frac{a}{x} < 0.
$$

Based on Bendixson–Dulac theorem (See [1]), we obtain that system (8) has no limit orbit in $R_2^+ = \{(x, y) | x \geq 0, y \geq 0\}$. The solution of system (7) is bounded in $R_2^+ = \{(x, y) | x \geq 0, y \geq 0\}$.

Then, if $(H)$ holds we are certain that the equilibrium point $E^*$ of system (7) is globally asymptotically stable. (As shown in Figure 1). □

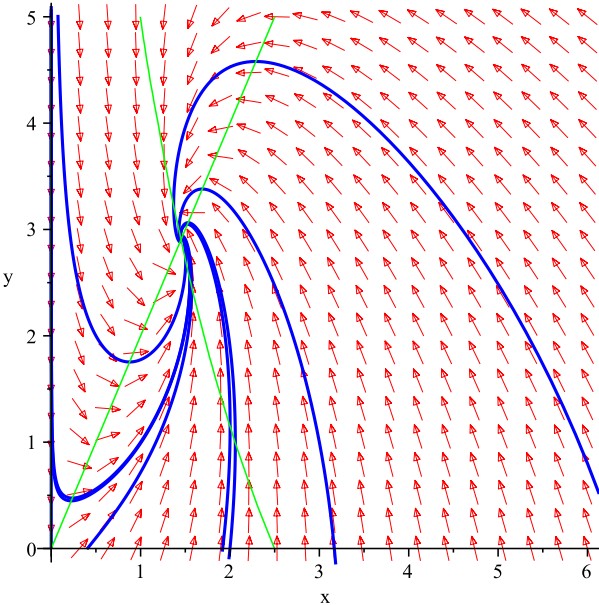

**Figure 1.** Phase diagram of system (7) with $r = 1.5, K = 2.5, D = 2, b = 0.1, a = 0.5$.

*2.2. Existence and Uniqueness of the Order-One Periodic Orbit of Impulsive State Feedback Control Model*

According to Theorem 1, we obtain that point $E^*$ is a stable node or focus and $O(0,0)$ is a saddle point, if $(H)$ holds. Then we discuss the existence and uniqueness of the OOPO of system (7) under this condition.

Let the impulsive set $M = \{(x,y) \in R_2^+ | x = h, y \geq 0\}$ and the phase set $N = \{(x,y) \in R_2^+ | x = (1-v)h, y \geq 0\}$ of system (7). By the ecological significance we get $0 < (1-v)h < h < K$. Then we get the following conclusions.

**Theorem 2.** *(1) If $0 < (1-v)h < h < x^*$ holds, system (7) has a unique OOPO.*
*(2) If $0 < (1-v)h < x^* < h < K$, then system (7) has no OOPO.*

**Proof.** (1) $0 < (1-v)h < h < x^*$.

The pulse set $M$ and phase set $N$ are on the left of the positive equilibrium point $E^*(x^*, y^*)$. The impulse set $M$ intersects with $x$-axis at point $M'(h, 0)$, and the phase set $N$ intersects with $x$-axis at point $N'((1-v)h, 0)$. The isoclinic $\dot{x} = 0$ intersects with phase set $N$ at point $B(h, y_B)$, where

$$y_B = \frac{r(K-h)}{b(K+Dh)}) > \frac{\sqrt{\Delta} - (ar + bK)}{2abD} = y^*.$$

Because the point $E^*$ is asymptotically stable, thus the orbit $f(B, t)$ will cut the pulse set $M$ at point $B_1(h, y_{B_1})$, then jumps to the point $B^1$ on the phase set $N$. The successor point $B^1$ of point $B$ is under the point $B$, that is to say,

$$y_{B^1} < \frac{r(K-h)}{b(K+Dh)},$$

then the successor function of point $B$ is

$$g(B) = y_{B^1} - y_B = y_{B^1} - \frac{r(K-h)}{b(K+Dh)} < 0.$$

The orbit $f(N', t)$ meets the impulsive set $M$ at point $A_1(h, y_{A_1})$, then jumps to the point $A^1((1-v)h, y_{A^1})$. So point $N'$ is under the subsequent point $A^1$ of point $A$, and

$$y_{A^1} > 0,$$

then the subsequent function of point $N'$ is

$$g(N') = y_{A^1} - y_{N'} = y_{A^1} > 0.$$

By [1], there must has a point $C$ between point $N'$ and $B$ such that

$$g(C) = 0,$$

then there exists an OOPO which crosses point $C$ in system (7). (See Figure 2)

Now, we prove the uniqueness of the OOPO of system (7).

First, we choose any point $J((1-v)h, y_J)$ and $I((1-v)h, y_I)$ such that

$$y_J > y_I.$$

The trajectory $f(J, t)$ will intersects with impulse set $M$ at point $J_1$, then jumps to the point $J^1$ by the impulse effect. The successor function of point $J$ is

$$g(J) = y_{J^1} - y_J.$$

The trajectory $f(I, t)$ will intersects with impulse set $M$ at point $I_1$, then jumps to the point $I^1$ by the impulse effect. The successor function of point $I$ is

$$g(I) = y_{I^1} - y_I.$$

We define the longitudinal coordinates of any point on the trajectory $f(P_0, t)$ to satisfy the function $y_{(x,P_0)}$, where $P_0$ is the starting point. Since $y_J > y_I$, then we define

$$y_J(x) = y_{(x,J)}, y_I = y_{(x,I)},$$

and

$$d_{JI} = y_J(x) - y_I(x), x \in [(1-v)h, h].$$

Thus, we have

$$
\begin{aligned}
d'_{JI}(x) &= y'_{(x,J)} - y'_{(x,I)} \\
&= \frac{K + Dx}{x} \left[ \frac{x - ay_J}{r(K-x) - by_J(K+Dx)} - \frac{x - ay_I}{r(K-x) - by_I(K+Dx)} \right] \\
&= \frac{K + Dx}{x} \varphi'(\zeta)(y_J - y_I) < 0,
\end{aligned}
$$

where

$$\varphi(y) = \frac{x - ay}{r(K-x) - by(K+Dx)},$$

with

$$
\begin{aligned}
\varphi'(y) &= \frac{-a[r(K-x) - by(K+Dx)] + b(K+Dx)(x-ay)}{[r(K-x) - by(K+Dx)]^2} \\
&= \frac{bDx^2 + bKx + arx - arK}{[r(K-x) - by(K+Dx)]^2} \\
&= \frac{bDx^2 + (bK+ar)x - arK}{[r(K-x) - by(K+Dx)]^2},
\end{aligned}
$$

when

$$
0 < (1-v)h < h < \frac{\sqrt{(bK+ar)^2 + arbD} - (bK+ar)}{bD},
$$

then

$$
\varphi'(y) < 0.
$$

Thus

$$
d'_{JI}(x) < 0
$$

for $x \in [(1-v)h, h]$. That is to say, $d_{JI}(x)$ is a decreasing function and

$$
d_{JI}(h) < d_{JI}((1-c)h).
$$

Let

$$
d_1 = d_{JI}((1-c)h) = y_J - y_I, \ d_2 = d_{JI}(h) = y_{J_1} - y_{I_1} = y_{J^1} - y_{I^1}.
$$

Obviously, $d_1 > d_2$. According to $y_I < y_J$, then $y_{I^1} < y_{J^1}$. Thus the subsequent function of point $I$ and $J$ satisfies

$$
\begin{aligned}
g(J) - g(I) &= (y_{J^1} - y_J) - (y_{I^1} - y_I) \\
&= -(y_J - y_I) + (y_{J^1} - y_{I^1}) \\
&= d_2 - d_1 < 0.
\end{aligned}
$$

That is to say, subsequent function $g(x)$ is monotonically decreasing in the phase set $N$, thus there must exist a unique point $S$ between point $A$ and $B$ such that $g(S) = 0$. It means the OOPO of system (7) is unique.

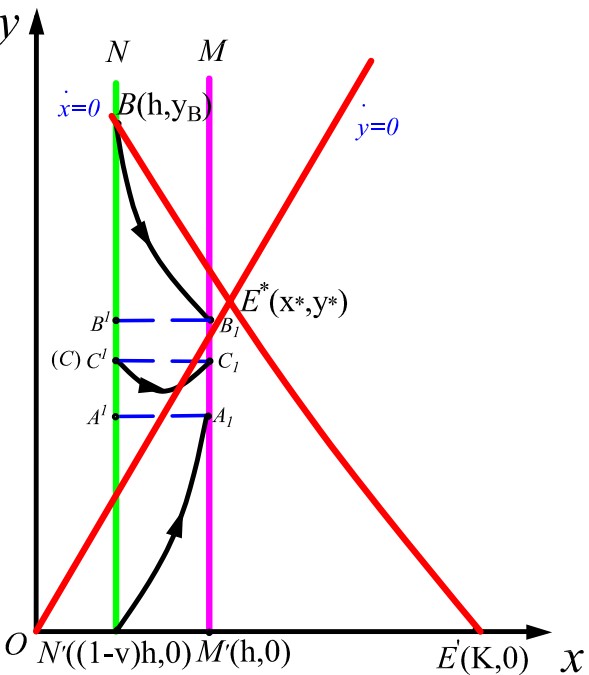

**Figure 2.** Existence and uniqueness of order-one periodic orbit if $0 < (1-v)h < h < x^* < K$ holds.

(2) If $0 < (1-v)h < x^* < h$.

The positive equilibrium point $E^*(x^*, y^*)$ is between at the phase set $N$ and pulse set $M$. The isoclinic line $\dot{x} = 0$ intersects with phase set $N$ at point $B[h, \frac{r(K-h)}{b(K+Dh)}]$. The pulse set $N$ intersects with the $x$-axis at point $N'((1-v)h, 0)$ and $M'(h, 0)$, respectively. According to the trajectory tend, the trajectory $f(B, t)$ will have no intersection point of impulse set $M$. If the trajectory $f(N', t)$ intersects with impulse set $M$ at point $Q_1(h, y_{Q_1})$, then jumps to the point $Q^1((1-v)h, y_{Q^1})$. According to the stability of the positive equilibrium point $E^*(x^*, y^*)$), the subsequent point $Q^1$ of point $N'$ must above the point $N'$ and $y_{Q^1} > 0$, that is to say, $g(N') > 0$. The trajectory $f(Q^1, t)$ will intersects with impulse set $M$ at point $Q_2$, then jumps to point $Q^2$, and point $Q^2$ surely above the point $Q^1$ such that $g(Q^1) > 0$. Repeat the above process, trajectory $f(Q^k, t)$ where $k = 1, 2, \cdots$ will have no intersection point of impulse set $M$. So system (7) has no OOPO (see Figure 3a).

If the trajectory $f(N', t)$ and pulse set $M$ are tangent to point $P_1$, then jumps to the point $P^1$. It is easy to know the subsequent point $P^1$ of point $N'$ is above the point $N'$. The subsequent function of point $N'$ is $g(N') = y_{P^1} - y_{N'} > 0$. The trajectory $f(P^1, t)$ will has no intersection point of impulse set $M$(See Figure 3b). If the trajectory $f(N', t)$ has no intersection point with the impulse set $M$, obviously, system (7) has no OOPO (See Figure 3c). Thus, we assume the region $G^*$ consists of one line segment $\overline{N'D}$, one curve $\widehat{DE*}$ and the trajectory $f(N', t)$.

When the trajectory starts from any point inside the region $G^*$, then the trajectory will tend to the positive equilibrium $E^*(x^*, y^*)$ by impulse effect. The trajectory starting from any point outside the region $G^*$, the trajectory also enter the region $G^*$ by several impulsive effects at most. Thus all trajectory will be attractive to the positive equilibrium $E^*$. Due to the discussion above, system (7) has no OOPO, when $0 < (1-v)h < x^* < h$.

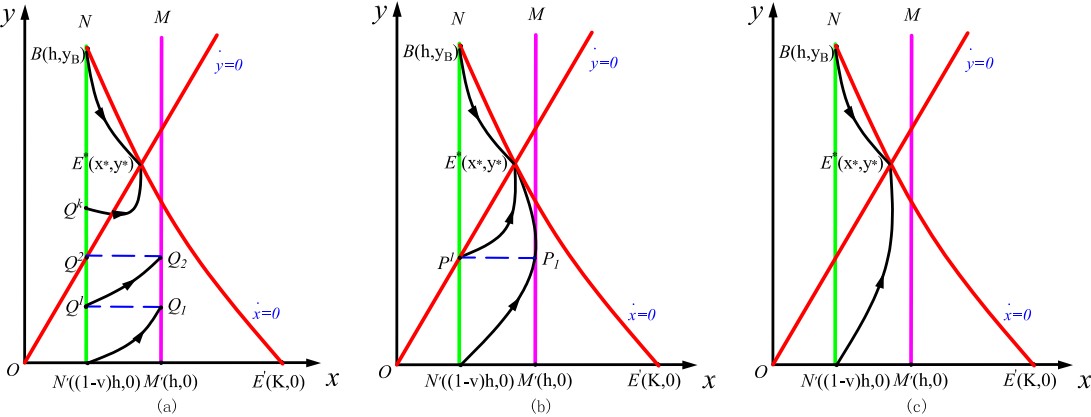

**Figure 3.** System (7) has no order-one periodic orbit if $0 < (1-v)h < x^* < h$ holds. (**a**) The trajectory $f(N',t)$ intersects with impulse set $M$; (**b**) the trajectory $f(N',t)$ and impulsive set $M$ are tangent; (**c**) the trajectory $f(N',t)$ will have no intersection point with impulse set $M$.

□

### 2.3. Stability of the Order-One Periodic Orbit of Impulsive State Feedback Control Model

**Theorem 3.** *The order-one periodic orbit of system* (7) *is stable.*

**Proof.** Assume that the intersection point of phase set $N$ and $\Gamma_1 = \dot{y} = 0$ is $F$ and impulsive set $M$ intersect $\Gamma_1$ at point $H$. By the pulse action, it can transfer the point $H \in M$ into the $J \in N$, one has

$$y_F = \frac{(1-v)h}{a}, y_J = y_H = \frac{h}{a}.$$

According to Theorem 2, we obtain that a point $S$ is in the straight line segment $\overline{JF}$, and the order-one periodic orbit will passing through this point, denoted by $f(S,t)$. It is easy to see, that $y_F < y_S < y_J$. Thus $g(F) > 0$, for any $F \in N$ with $y_S > y_F$, $g(J) < 0$ for any $J \in N$ with $y_J > y_S$ and $g(F) = g(J) = 0$ if and only if $J = F = S$. Then we choose any point $F^0 \in N$. If $F^0 \in N/\overline{FS}$, then after several impulsive effects, the trajectory will jump to $\overline{FS}$. Thus assume that $F^0 \in \overline{FS} \subset N$, the trajectory $f(F^0,t)$ intersects with impulse set $M$ at point $F_1$, then the point $F_1$ becomes the point $F^1 \in N$ after impulse effect and $y_F < y_{F^0} < y_{F_1} = y_{F^1}$, one has

$$\frac{(1-v)h}{a} = y_F < y_{F^0} < y_{F^1} < y_s.$$

The trajectory $f(F^1,t)$ will intersect with impulse set $M$ at point $F_2$, and $y_F < y_{F^0} < t_{F^1} < y_{F^2} < y_s$. Repeat the above process, we can obtain the point sequences $\{F^k\}$, where $k = 0,1,2,\cdots$ such that

$$\frac{(1-v)h}{a} = y_F < y_{F^0} < y_{F^1} < \cdots < y_{F^k} < \cdots \leq y_s.$$

Then the sequence $F^k \parallel_{k=0,1,2,\cdots}$ is a monotonic increasing sequence with upper bound $y_s$. By the monotone bounded theorem, there must has a limit $y_{S'}$ such that $\lim_{k\to\infty} y_{F^k} = y_{S'}$, which implies that

$$g(S') = g\left(\lim_{k\to\infty} y_{F^k}\right) = \lim_{k\to\infty} g(y_{F^k}) = \lim_{k\to\infty} (y_{F^{k+1}} - y_{F^k}) = 0$$

Since $g(F) = 0$, if and only if $F = S$, then $S' = S$. That is to say, $\lim_{k\to\infty} g(y_{F^k}) = y_{S'} = y_S$.

Similarly, we also choose any point $J^0 \in N$. If $J^0 \in N/\overline{JS}$, after many times impulsive effect the trajectory will jump to $\overline{JS}$. So we assume that $J^0 \in \overline{JS} \subset N$, the intersection point of the trajectory $f(J^0,t)$ and impulse set $M$ is $J_1$, then jumps to the point $J^1$, and $y_S < y_{J^1} < y_J$. Then the trajectory

$f(J^1, t)$ intersects with impulse set $M$ at point $J_2$, then jumps to the point $J^2$. Repeating the above process, we get the point sequence $\{J^k\}$, where $k = 0, 1, 2, \cdots$ such that

$$y_S \leq \cdots < y_{J^k} < \cdots < y_{J^2} < y_{J^1} < y_{J^0} < y_J = \frac{h}{a}.$$

Since $\{J^k\}$ is a monotonic decreasing sequence with lower bound $y_S$. By the monotonic bounded theorem, there must be a limit $y_{S'}$ such that $\lim_{k\to\infty} y_{J^k} = y_{S'}$, which means that

$$g(S') = g\left(\lim_{k\to\infty} y_{J^k}\right) = \lim_{k\to\infty} g(y_{J^K}) = \lim_{k\to\infty} (y_{J^{k+1}} - y_{J^k}) = 0.$$

Since $g(J) = 0$, if and only if $J = S$, then $S' = S$. That is to say, $y_{J^k} = y_S$. Therefore, we have

$$\frac{(1-v)h}{a} = y_F < y_{F^0} < y_{F^1} < \cdots < y_{F^k} < \cdots \leq y_S \leq \cdots < y_{J^k} < \cdots < y_{J^2} < y_{J^1} < y_{J^0} < y_J = \frac{h}{a}.$$

According to the arbitrariness of the point $F^0$ and $J^0$, we have

$$\lim_{k\to\infty} y_{F^k} = \lim_{k\to\infty} y_{J^k} = y_S.$$

That is to say, all trajectories of system (7) tend to the OOPO (see, e.g., [50]). Thus the OOPO of system (7) is orbitally asymptotically stable and globally attractive. (See Figure 4).

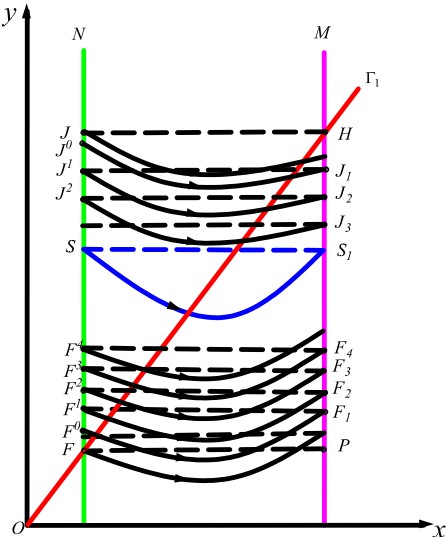

**Figure 4.** The orbitally asymptotically stable and global attractive of order-one periodic orbit of system (7).

□

## 3. Numerical Simulations and Conclusions

### 3.1. Numerical Simulations

In the section above, we have proved existence, uniqueness and stability of OOPO of system (7). In this section, theoretical results are verified by numerical simulation results.

Let $r = 1.5, K = 2.5, D = 2, b = 0.1$ and $a = 0.5$. By calculations, we get that the positive equilibria is $E^*(1.4528, 2.9056)$. Meanwhile, $\Delta = (ar + bk)^2 + 4abrDK = 2.5 > 0, K < x^*\sqrt{D} = 2.51$. Then the system (7) becomes

$$\begin{cases} \left. \begin{array}{l} \dfrac{dx(t)}{dt} = x\left(\dfrac{1.5(2.5 - x)}{2.5 + 2x} - 0.1y\right), \\[2mm] \dfrac{dy(t)}{dt} = x - 0.5y, \end{array} \right\} \quad x < h, \\[4mm] \left. \begin{array}{l} \triangle x(t) = -vx, \\[1mm] \triangle y(t) = 0, \end{array} \right\} \quad x = h. \end{cases} \tag{10}$$

Case I: Let $h = 1, v = 0.5$, then $0 < (1 - v)h < h < x^*$. Figure 5. shows that an OOPO, which starts from initial point $(0.5, 1.56)$ exists in system (7).

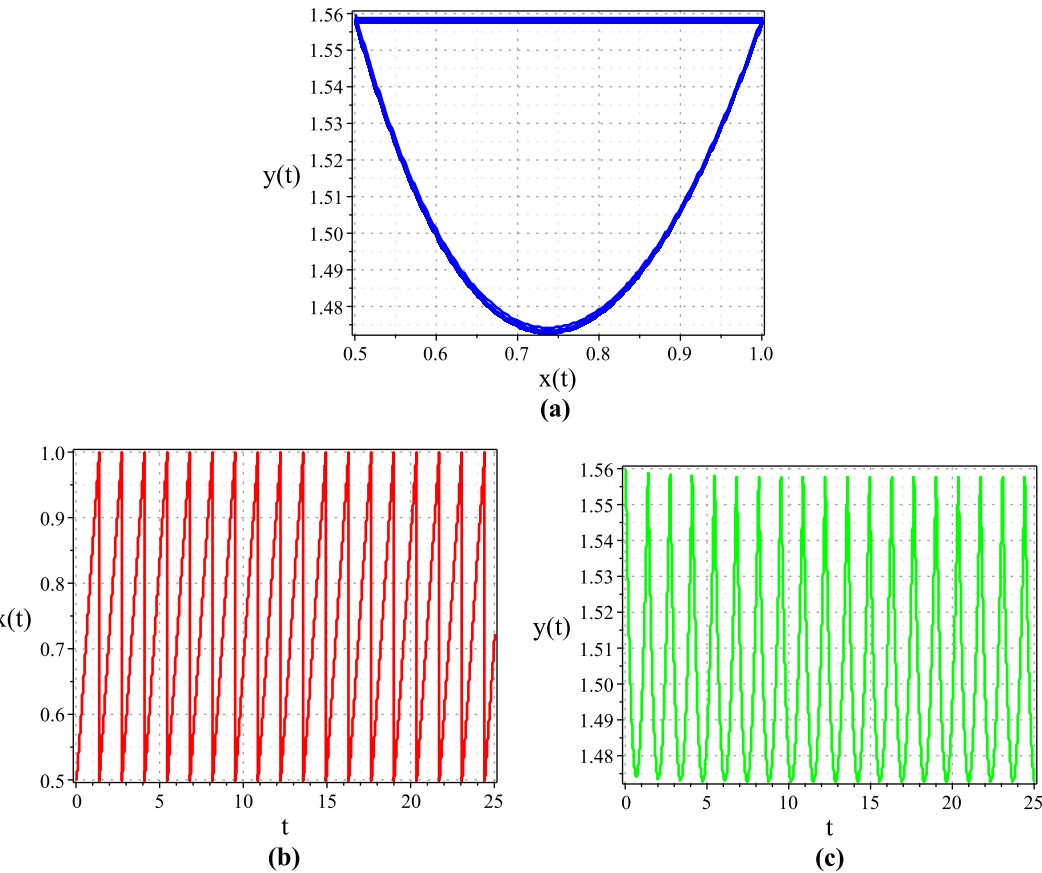

**Figure 5.** Phase portrait and time series of system (7) with $h = 1, v = 0.5$. (**a**) Phase portrait of x(t) and y(t). (**b**) Time series of x(t). (**c**) Time series of y(t).

Case II: Let $h = 1.5, v = 0.5$, then $0 < (1 - v)h < x^* < h$. Figure 6. shows that an OOPO which starting from initial point $(0.5, 1.56)$ does not exist in system (7).

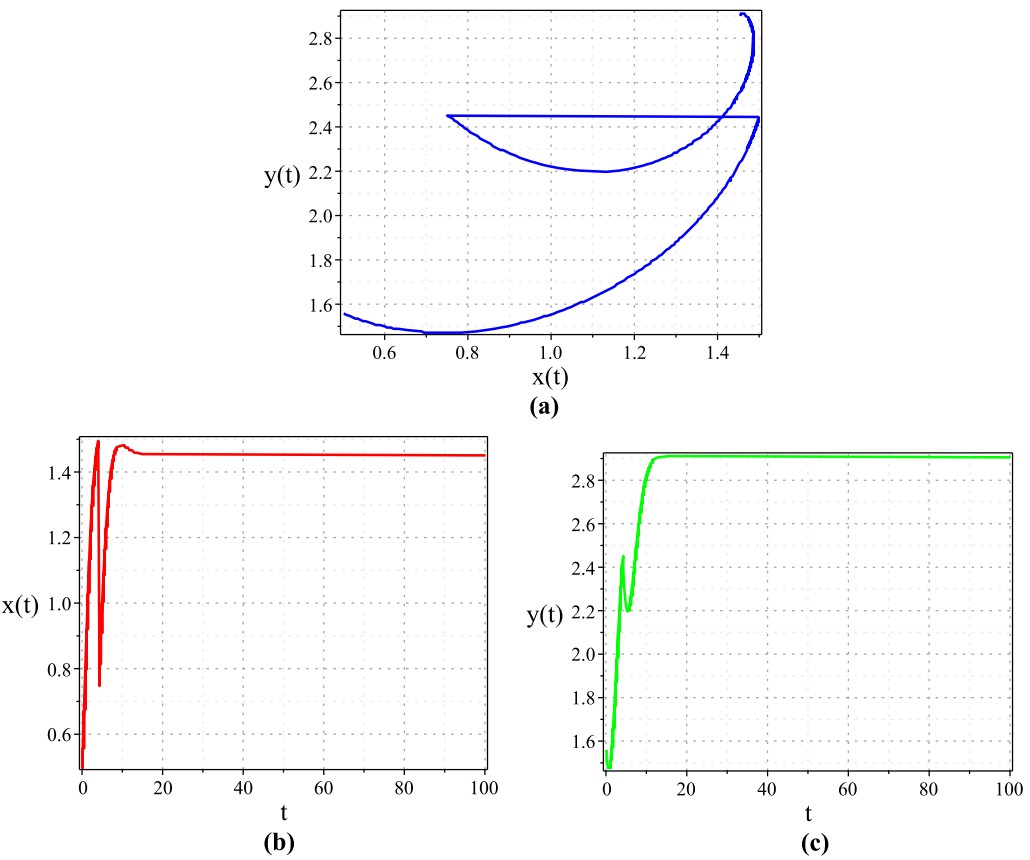

**Figure 6.** Phase portrait and time series of system (7) with $h = 1.5, v = 0.5$. (**a**) Phase portrait of x(t) and y(t). (**b**) Time series of x(t). (**c**) Time series of y(t).

We assume $r = 1.5, K = 25, D = 2, b = 0.1, a = 0.5, h = 4, v = 0.5$ and pest initial value $x(t) = 2$. Figure 7 shows that the pest population can be effectively controlled by the feedback control strategy.

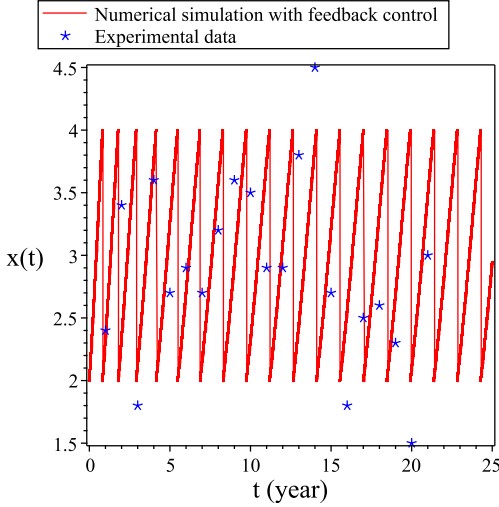

**Figure 7.** The parameter x denotes the density of pests. The red line denotes the sample path of pest species with continuous delay and feedback control and the blue star denotes the experimental data (the data comes from the Wanfang data knowledge service platform).

### 3.2. Conclusions

In this paper, we studied a pest management Smith model with impulsive state feedback control and continuous time delay. By the feedback information of the density of forest pest population from the monitor, we can effectively protect the forest ecosystem.

First, we carried on the quantitative and qualitative analysis, and obtained a condition $(H)$. We proved the global asymptotical stability of the positive equilibrium $E^*(x^*, y^*)$ by the theory of Bendixson–Dulac.

Then we proved the existence of the OOPO of system (7) by the geometric theory of differential equations, and the uniqueness of the OOPO of system (7) was proved by the monotonicity of the successor functions and Lagrange mean value theorem.

Finally, the orbital asymptotical stability of the OOPO was studied by the geometry properties of successor functions. Then we proved the existence of the limitation by the uniqueness of the OOPO and limit existence theorem.

All the results suggest that the number of pests will be finally controlled to be an OOPO. By the different plant growth cycle, the detection of pest number can help people to manage the density of pests, so as to protect ecological environment.

**Author Contributions:** The main idea of this paper was proposed by H.C., Y.L. was responsible for completing the first draft of the paper and Z.S. is responsible for the revision of the paper and the final proofreading. Furthermore, all authors read and approved the final manuscript.

**Funding:** The paper was supported by the National Natural Science Foundation of China (No. 11371230), Shandong Provincial Natural Science Foundation of China (No. ZR2019MA003), SDUST Research Fund (2014TDJH102), and Joint Innovative Center for Safe and Effective Mining Technology and Equipment of Coal Resources, Shandong Province of China.

**Conflicts of Interest:** The authors declare no conflict of interest.

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
