# Peer review of "Dynamic Analysis of a Pest Management Smith Model with Impulsive State Feedback Control and Continuous Delay"

_mathematics, doi:10.3390/math7070591_

Round 1

Reviewer 1 Report

This is a nice paper that was easy to follow and I have no major criticisms. I have a number of points that need clarification though:

In equation (2) you introduce the terms \delta x(t), v and h without explanation. Can you add some text explaining what they are. (I was only able to deuce their meaning from the context later in your paper).

On line 55, I could not understand what you are saying in the sentence starting with Let (H): K < ... Could you possibly make it a bit clearer.

On line 57, the Jacobian at O(0,0) should have r as the 0,0 element (unless I'm mistaken the Jacobian gives rK^2/K^2 at 0,0).

On line 77, you use a superscript 1 and subscript 1 to denote successor points. Do you mean to do this? I assume you do as you use the same notation later. However, at the end of the second paragraph of the proof you omit the subscript on B - is this correct?

Finally, you use the term OOPO in several places, please add it on line 70 as

"... and uniqueness of the order-one periodic orbit (OOPO) of system ..."

There are several grammatical errors in the translation that the publication system will, I'm sure, pick up.

Author Response

Response to Reviewer 1 Comments

Manuscript number: mathematics-503848

Title: Dynamic analysis of a pest management Smith model with impulsive

state feedback control and continuous delay

This is a nice paper that was easy to follow and I have no major criticisms. I have a number of points that need clarification though:

Response 1:We thank you for your careful reading of the original manuscript, and valuable comments and suggestions for improving the quality of the paper. Thank you so much for giving us a chance to revise the paper.

Point 1: In equation (2) you introduce the terms Δx(t), v and h without explanation. Can you add some text explaining what they are. (I was only able to deuce their meaning from the context later in your paper).

Response 1: Thanks. △x(t) = x+(t) − x(t), i.e △x(t) represents the change in population density of pests. v is proportion of killing pests by spraying insecticides. h denotes the economic threshold of pests, i.e. the control measure is adopted when pest population density is above h. We also made changes in the manuscript. Please see Line 39, 40 and 41.

Point 2: On line 55, I could not understand what you are saying in the sentence starting with Let (H) : K < ::: Could you possibly make it a bit clearer.

Response 2: Thanks. We have made further explanation. Please see Line 57.

Point 3: On line 57, the Jacobian at O(0,0) should have r as the 0,0 element (unless I’m mistaken the Jacobian gives rK2=K2 at 0,0).

Response 3: Thanks. We have made the change. Please see Line 59.

Point 4: On line 77, you use a superscript 1 and subscript 1 to denote successor points. Do you mean to do this? I assume you do as you use the same notation later. However, at the end of the second paragraph of the proof you omit the subscript on B - is this correct?

Response 4: Thanks. We use a superscript 1 to denote successor points and subscript 1 to denote the intersection of the trajectory and the impulse set. Through careful inspection, we do not find B that is omitted subscript 1.

Point 5: Finally, you use the term OOPO in several places, please add it on line 70 as ”... and uniqueness of the order-one periodic orbit (OOPO) of system ...”.

Response 5: Thanks. We have added term OOPO. Please see Line 73.

Point 6: There are several grammatical errors in the translation that the publication system will, I’m sure, pick up. Response 6: Thanks. According to this comment, we have checked the English carefully and corrected grammar errors. We have already marked it in blue in the manuscript.

Reviewer 2 Report

I am attaching comments.

Author Response

We thank you for your careful reading of the original manuscript, and valuable comments and suggestions for improving the quality of the paper. Thank you so much for giving us a chance to revise the paper.  Please see PDF for specific response.

Reviewer 3 Report

The article deals with the study of an impulsive control of a Smith model. Particularly, the dynamical system with impulse is investigated for existence and uniqueness for one-order periodic orbits. The authors try to link this model to pest management in forests. After a theoretical analysis of the system, a small numerical section is enclosed.

The English used throughout the article is rather poor. I recommend using one a spell and grammar check before resubmitting this work. The presentation is also to improve. For example assumptions like (H) are hidden in the text. They should be placed such that they can be found easier. 

In my opinion, the relation to the application is unclear. Especially the consequences of this study concerning the applications are not given. The analysis of the system is straight forward and I think it is rather an easy example for a teaching book than a research article. 

The proofs, however, seem to be correct. 

I hence recommend rejecting the article in its current state due to lack of originality.

Author Response

Response to Reviewer 3 Comments

Manuscript number: mathematics-503848

Title: Dynamic analysis of a pest management Smith model with impulsive state feedback control

and continuous delay

Point 1: The article deals with the study of an impulsive control of a Smith model. Particularly, the

dynamical system with impulse is investigated for existence and uniqueness for one-order periodic

orbits. The authors try to link this model to pest management in forests. After a theoretical analysis

of the system, a small numerical section is enclosed.

Response 1: We thank you for reading the original manuscript carefully and providing valuable

comments and suggestions for improving the quality of the paper.

Point 2: The English used throughout the article is rather poor. I recommend using one a spell and

grammar check before resubmitting this work.

Response 2: Thanks. We have checked and corrected the spelling and grammar errors of the full

text. We have already marked it in blue in the manuscript.

Point 3: The presentation is also to improve. For example assumptions like (H) are hidden in the

text. They should be placed such that they can be found easier.

Response 3: Thanks. We checked the presentation of the entire manuscript. And we put the hidden

assumptions in a location that is easier to find. Please see Line 57.

Point 4: In my opinion, the relation to the application is unclear. Especially the consequences of

this study concerning the applications are not given. The analysis of the system is straight forward

and I think it is rather an easy example for a teaching book than a research article.

Response 4: Thank you for your comment. We assume r = 1:5;K = 25;D = 2; b = 0:1; a =

0:5; h = 4; v = 0:5 and pest initial value x(t) = 2. Figure 7 shows that the pest population can be

effectively controlled by the feedback control strategy. Therefore, Through specific data, our control

strategy has practical significance.

Round 2

Reviewer 2 Report

The authors use a single-species model to represent a pest species and investigate the dynamics when it is subjected to control by periodic pulses. They prove the existence, uniqueness, and stability of an order-one periodic orbit (OOPO), which they then support by some numerical simulations.

Comments

I reviewed an earlier version of this manuscript.  The authors have responded satisfactorily to my comments. In particular, the transition from equation (4) to equation (6) seems reasonable to me; the additional term in equation (6) represents mortality rate with a continuous time delay. I think the paper is a useful contribution to theory of pest management.

Reviewer 3 Report

Most of my comments were treated. The manuscript is publishable.